# Use of Selected Lactic Acid Bacteria for the Fermentation of Legume-Based Water Extracts

**DOI:** 10.3390/foods11213346

**Published:** 2022-10-25

**Authors:** Chiara Demarinis, Michela Verni, Loris Pinto, Carlo Giuseppe Rizzello, Federico Baruzzi

**Affiliations:** 1Department of Soil, Plant and Food Sciences, University of Bari Aldo Moro, 70126 Bari, Italy; 2Institute of Sciences of Food Production, National Research Council of Italy (CNR-ISPA), 70126 Bari, Italy; 3Department of Environmental Biology, Sapienza University of Rome, Piazzale Aldo Moro 5, 00185 Rome, Italy

**Keywords:** lactic acid bacteria, legumes, bioprocessing, non-dairy alternatives, sensory properties, free amino acids and peptides

## Abstract

In this study, the effect of selected *Lactobacillus acidophilus* ATCC 4356, *Limosilactobacillus fermentum* DSM 20052, and *Lacticaseibacillus paracasei* subsp. *paracasei* DSM 20312 strains on the sensory characteristics, and protein and amino acid content of fermented water extracts derived from lupin, pea, and bean grains is reported. Even though all strains were able to grow over 7 log cfu mL^−1^ and to decrease pH in the range of −0.52 to −1.25 within 24 h, the release of an unpleasant ferric-sulfurous off-odor from the fermented bean water extract prohibited further characterization. Lupin and pea grain-based beverages underwent an in-depth sensory evaluation using a simplified check-all-that-apply (CATA) method, finding new and appreciable sensory notes such as cooked ham, almonds, and sandalwood. Fermented lupin water extract showed higher total protein content (on average, 0.93 mg mL^−1^) in comparison to that of pea grains (on average, 0.08 mg mL^−1^), and a free amino acid content (on average, 3.9 mg mL^−1^) close to that of cow milk. The concentrations of these nutrients decreased during refrigerated storage, when the lactic acid bacteria load was always higher than 7 log cfu mL^−1^. The results of this study indicated that lactic fermentation improves the sensory characteristics of these innovative legume-based beverages, which sustained high loads of viable lactobacilli up to the end of cold storage.

## 1. Introduction

Although milk is essential for the body development of mammals, thanks to its content of high-biological-value nutrients, its consumption is decreasing due to lactose intolerance, cow’s milk protein allergy, and to the increase in people following a vegan or vegetarian diet [1]. In addition, part of the reduction in milk and dairy product consumption arises from a new consumer awareness towards animal welfare [2] and the reduction of the carbon footprint of animal farms [3]. Consequently, the world demand for non-dairy alternatives is estimated to grow by around 250% in the period 2020–2028 [4]. 

Plant-based products can be a good alternative. In agreement with Chaturvedi and Chakraborty [5], plant-based milk substitutes can be described as water-soluble extracts of legumes, oilseeds, cereals, or pseudo-cereals that resemble bovine milk in appearance and can be produced by different methods. Lactic acid bacteria (LAB) fermentation of legume grains, flours, and extracts is a sustainable approach to increase their (i) mineral bioavailability thanks to microbial and/or endogenous phytases, able to hydrolyze phytic salt into free inositol, metal ions (mainly calcium, iron, magnesium, and zinc) and phosphates, and (ii) protein and amino acid availability resulting from microbial proteolytic activity against structural seed proteins and specific enzymes responsible for the inhibition of trypsin and chymotrypsin under gastric environmental conditions [6,7]. In addition, LAB decrease the concentration of fermentable oligo-, di-, monosaccharides, and polyols in fermented legumes, with particular concern regarding the raffinose family of oligosaccharides, considered anti-nutritional factors responsible for flatulence [6,7].

Soy (*Glycine max*, L.) grains are the most studied and employed legumes for the production of fermented plant-based beverages, and the effect of lactic acid fermentation on the quality of soy milk was reported more than 40 years ago [8]. However, other legume grains, such as lupins [9], beans [10,11,12], chickpeas [13,14,15], or faba beans [16], have been also fermented by LAB. The characterization of a novel lentil-based beverage fermented with different LAB strains showed a strong reduction in both phytic acid and raffinose oligosaccharide, well known as anti-nutritional factors, and a high concentration of soluble and highly digestible proteins [17]. In addition, a yogurt-like beverage including chickpea and lentil flours has been fermented with *Lactobacillus* strains, increasing the antioxidant activity and sustaining the survival of a commercial probiotic strain [18].

Recently, the application of different technologies allowed the realization of new beverages from pea, chickpea, and lupin, incorporating a large amount of seed components with low release of by-products [19]. 

It is well known that the sensory profiles of legume-based beverages are the main key restraints that prevent their larger diffusion among consumers. In particular, volatile compounds such as n-hexanal and n-hexanol, which originate from lipid oxidation, are mainly responsible for the beany off-flavor [20], whereas tannins and saponins, terpenes, glucosinolates, and flavonoids impart bitter or astringent tastes [21]. In addition, greenish, greyish, or brownish colors, and a chalky or sandy texture, due to the presence of insoluble particles, are the main sensory features negatively affecting the consumer acceptability of legume-based beverages [20]. 

Some of these drawbacks can be reduced by LAB fermentation thanks to the production of several metabolites, such as lactic and acetic acid or acetoin and acetaldehyde, able to reduce the beany flavor, to enhance the fruity flavor, or to mask the “green-note” off-flavor [22,23,24]. However, their excess concentration, as demonstrated for acetic acid in beer, could also be detrimental to the organoleptic acceptance of the beverage [25].

Thus, the aim of this work was to ferment different legume grain watery extracts to obtain a legume-based beverage with acceptable sensory traits, rich in soluble peptides and amino acids, that could be a source of probiotic lactobacilli at the end of the cold storage period. 

## 2. Materials and Methods

### 2.1. Legume-Based Water Extract Preparation and Enumeration of Their Autochthonous Microbial Populations

Pea (*Pisum sativum* Asch. et Gr.), bean (*Phaseolus vulgaris* L.), and lupin (*Lupinus albus* L.) were kindly provided by Terre di Altamura S.r.l. (Altamura, Bari, Italy) and used to prepare legume-based fermented beverages. 

Grains were dipped in tap water for around 16 h using the ratio of dry grain/water of 1:10 (*w*/*w*). Since grains absorbed different amounts of water, the ratio of dry grain/water was maintained at 1:10 by adding fresh water to soaked grains. Grain suspensions were homogenized by a hand blender and filtered using cotton gauze [17]. 

Legume- based water extracts were characterized for their main microbial populations before and after incubation at 24 h at 37 °C. The extracts were serially diluted in sterile 0.1% *w*/*v* buffered peptone water. Then, dilutions were plated in triplicate on (i) Plate Count Agar (PCA) for counting of total aerobic bacteria (24 h at 30 °C, as required by the ISO standard n° 4833 [26]), (ii) acidified (pH 5.4) de Man, Rogosa, and Sharpe (MRS Agar ISO formulation) for mesophilic lactic acid bacteria (anaerobic incubation for 48 h at 30 °C, as already reported [27]); (iii) Potato Dextrose Agar (PDA) supplemented with chloramphenicol (0.1 g/L) for yeasts and molds (incubation for 3 days at 25 °C), and (iv) Reinforced Clostridial Medium (RCM) immediately after thermal treatment (80 °C for 10 min) for aerobic spore-forming bacteria (incubation at 30 °C for 3 days [28]). 

All media were purchased from Biolife Italiana S.r.l., Milan, Italy. Microbial loads were expressed as log colony forming unit (cfu) mL^−1^.

### 2.2. Lactic Acid Fermentation of Legume-Based Water Extracts

Strains of the former *Lactobacillus* genus used in this work were included in the Agro-Food Microbial Culture Collection (ITEM) at the Institute of Sciences of Food Production of Bari, Italy (http://server.ispa.cnr.it/ITEM/Collection/ (accessed on 13 September 2022) (Table 1). Fresh microbial cultures of *Lactobacillus* spp. strains from frozen cultures (−80 °C) were routinely grown in MRS (MRS Broth ISO Formulation, Biolife Italiana S.r.l., Milan, Italy) for 48 h at 37 °C under anaerobic conditions (Anaerogen, AN0025, Oxoid S.p.A., Milan, Italy). 

When used for fermentation, fresh lactobacilli cultures were centrifuged (10,000 rpm for 3 min, centrifuge model Sigma 3–30 KS, Sigma Laborzentrifugen GmbH, Germany), removing the supernatant and re-suspending the cell pellet in saline solution until an absorbance reading at 600 nm of 0.3 ± 0.05 (ca. 8 log cfu mL^−1^) was obtained. Before lactic acid fermentation, legume-based water extracts were sterilized in an autoclave at 110 °C for 10 min, as previously reported [17]. The three sterilized legume-based water extracts were singly inoculated with 1% of cell suspension (final cell density of around 5–6 log cfu mL^−1^) and incubated for up to 48 h at 37 °C under anaerobiosis [17].

Before and after fermentation, viable cell counting of lactic acid bacteria was carried out as reported in Section 2.1, incubating plates at 37 °C under anaerobiosis, and the pH was measured (Model pH50 Lab pH Meter XS-Instrument, Concordia, Italy). In order to understand the ability of different strains to grow in legume-based water extracts, as well as their ability to acidify them, the values of Δlog cfu mL^−1^ and ΔpH were calculated. These differences were achieved by subtracting the values obtained at 24 h or 48 h from those of the previous sampling time, e.g., the beginning of fermentation or 24 h.

### 2.3. Microbial, Biochemical, and Nutritional Characterization of the Legume-Based Beverages

#### 2.3.1. Shelf-Life Evaluation

Legume-based beverages, fermented with the three best-fermenting *Lactobacillus* strains selected during fermentation assays, were stored at 4 °C for 28 days. Changes in microbial populations and acidification rate capabilities were evaluated during cold storage (4 °C) every 7 days.

#### 2.3.2. Total Proteins, Peptides, and Free Amino Acids (FAA)

Total protein content and concentrations of peptides and free amino acids were analyzed at 7-day intervals during cold storage. The protein concentration was evaluated using the Bradford method [29]. Standard solutions of bovine serum albumin (BSA, Sigma No. A-7030) in the range of 0.05–1 mg mL^−1^ were used to build the calibration curve. Subsequently, the absorbance of unknown samples was measured at 595 nm by an automatic spectrofluorometer (Varioskan Flash, ThermoFischer Scientific, Milan, Italy) in multiwell plates. For each well, 5 readings were taken after incubation for 5 min at 25 °C and brief stirring (5 min at 120 rpm). 

The concentration of free peptides and free amino acids was performed through the o-phtaldialdehyde method (OPA) [30]. A standard curve of casamino acid mixture (BD BactoCasamino Acids, BD Biosciences, San Jose, CA, USA) was used as a reference in the range of 0.1–2.0 mg mL^−1^.

For the measurement of free amino acid concentrations, a Biochrom 30 series amino acid analyzer (Biochrom Ltd., Cambridge, UK) equipped with a Li cation exchange column (20 cm × 0.46 cm of internal diameter) was used [31]. 

#### 2.3.3. Sensory Analysis

The sensory analyses of the fermented legume-based beverages were carried out during 28 days of cold storage at regular 7-day intervals. The check-all-that-apply (CATA) method [32] was simplified to describe the visual, olfactory, and taste characteristics of the beverages. In particular, the group of panelists was composed of 10 untrained judges in the recognition of different sensory properties, as differently reported [33,34]. They evaluated the different fermented beverages, describing their main perceptions simply as they were perceived. 

Since these beverages were produced at lab scale for the first time, at the first instance, it was necessary to collect preliminary information about their acceptance. Thus, main traits were scored for the macro-descriptors “Appearance”, “Odor”, and “Taste” as 0—unexpected and/or unwanted feature, or 1—expected and/or welcome feature. The scores were then normalized for the number of panelists with the result of expressing the three macro-descriptors in a 0–1 interval. Even though this approach was less informative in comparison to other descriptive tests based on a hedonic scale for each descriptor, it was considered more useful to distinguish among samples by considering only the main, “good” or “bad”, trait perceived.

Participants were recruited among people answering a short survey sent by email to students, researchers, and professors, as well as members of their families, already in contact with the authors and their colleagues. Then, panelists were selected among those habitually consuming fermented beverages (different from wine and beer) and legumes. During the first session of sensory analysis, a moderator explained the aim of the study, the characteristics of the products (liquid and ready-to-drink), and instructions on how to complete the questionnaire, and also answered technical questions and queries. 

### 2.4. Statistical Analysis

The average lactic acid strain population enumerated during the cold refrigerated period for each legume-based beverage was analyzed by two-way analysis of variance (ANOVA); the significance of differences (*p* < 0.05) between mean values was evaluated by Fisher’s least significant difference test. Statistical differences in protein concentration and free amino acid and peptide concentration were evaluated by the T test. Statistical analyses were performed with Microsoft Excel software, implemented with the statistical analysis tool add-in (Microsoft Corporation, Redmond, WA, USA). 

## 3. Results and Discussion

### 3.1. Autochthonous Microbial Populations of Legume-Based Water Extracts

Viable cell counts of autochthonous microbial populations of legume-based water extracts, before and after 24 h of fermentation at 37 °C, are shown in Figure 1. The legume-based water extracts showed the presence of different types of autochthonous microbial populations, mainly belonging to presumptive lactic acid rods and cocci. In particular, LAB counts, presumptively represented by lactic acid rods, ranged from 2.39 ± 0.36 log cfu mL^−1^ to 4.62 ± 0.22 log cfu mL^−1^, with lupin and peas having the lowest and highest density, respectively (Figure 1). Aerobic endospore-forming bacteria were found before incubation only in peas and beans. However, after the incubation, they were detected in all samples, reaching 5.85 ± 0.01 log cfu mL^−1^ in beans, 6.10 ± 0.09 log cfu mL^−1^ in lupin, and approximately 1.88 ± 0.09 cfu mL^−1^ in pea water extracts (Figure 1). Yeast and molds were found only in the bean extracts, but, after incubation, they were not further detected (data not shown). 

The comparison of the means of each population before and after incubation within the same extract resulted in statistically significant values for all microbial populations, as shown by the asterisks in Figure 1, excluding aerobic endospore-forming bacteria of peas. 

These results confirmed that the legume grains were characterized by autochthonous and potentially useful lactic acid bacteria, able to grow during incubation, and spoilage bacteria, such as the endospore-forming ones. 

As far as the lactic acid microbial population is concerned, the viable loads here reported are in agreement with those found in several fruit and vegetables [35,36,37,38], belonging to species such as *Lactiplantibacillus plantarum*, *Lactiplantibacillus pentosus*, *Lm. fermentum*, *Latilactobacillus curvatus*, *Levilactobacillus brevis*, *Leuconostoc mesenteroides*, *Weissella* spp., and *Enterococcus* spp. 

As found for lactic acid bacteria, endospore-forming bacterial loads also increased during incubation. This result, in agreement with the occurrence of *Bacillus cereus, B. nitratireducens, B. pumilus, B. safensis,* and *B. australimaris* in different types of fermented food [39,40], forced us to heat-treat extracts before fermentation. 

### 3.2. Fermentation Assays of Legume-Based Water Extracts

In order to select the LAB strains for the production of fermented legume-based beverages, legume-based water extracts, after sterilization (110 °C for 10 min), were first inoculated with each strain at 5–6 log cfu mL^−1^ and incubated for 48 h at 37 °C. As shown in Table 2 and Table 3, *L. acidophilus* ATCC 4356, *Lm. fermentum* DSM 20052, and *Lc. paracasei* DSM 20312 showed the highest increase in average viable cell counts and a marked decrease in pH values within 48 h of fermentation in comparison to the beginning of incubation. 

All LAB strains grew in legume extracts, but only *L. acidophilus* ATCC 4356, *Lm. fermentum* DSM 20052, and *Lc. paracasei* DSM 20312 showed Δlog cfu mL^−1^ higher than one in their viable loads in the three legume extracts after 48 h at 37 °C (Table 2). 

The fermentation assays employing these strains were repeated, lowering the initial inoculum level to 4–5 log cfu mL^−1^ and evaluating pH changes and cell viability at 24 h and 48 h (Table 4 and Table 5).

Two-way ANOVA analysis showed that legume-based beverage, starter strain, and their interaction significantly (*p* ≤ 0.05) affected cell density and pH values at each sampling time. The effect of the strain on cell density values was not significant at 48 h. All strains confirmed their ability to grow in all legume-based water extracts already after 24 h, without excessive acidification, except in the case of *L. acidophilus* ATCC 4356 in the pea extract. After 48 h of incubation, the pea extract showed the lowest viable cell increase. The same legume-based beverage showed the highest reduction in pH values for all three inoculated strains after 48 h of incubation. Generally, the pH decreased differently during fermentation, depending on the legume extract and the strain used. 

The extension in incubation time from 24 to 48 h increased the average viable cell count only by 0.53 Δlog cfu mL^−1^ (as calculated from values reported in Table 4), with a drop in the pH values from 5.57 ± 0.49 to 4.76 ± 0.19 (Table 5). Results confirmed the ability of these strains to grow and reduce the pH in fermented legumes, as reported in soy for *L. acidophilus* ATCC 4356 [41], and for these three strains in lentil grains [17].

Thus, the fermentation step of the legume-based beverage for these strains was set up following incubation for 24 h at 37 °C, avoiding the increase in the concentration of acids from sugar fermentation, which could be unpleasant for the consumer.

As far as the bean water extract is concerned, the release of off-odors at the end of fermentation, probably due to volatile sulfur compounds, prevented its use for the production of a bean-based fermented beverage. As a consequence, bean water extracts were not further characterized.

### 3.3. Evaluation of the Shelf-Life during Cold Storage

Based on previous results, the fermented legume-based beverages were prepared from lupin and pea water extracts singly fermented for 24 h at 37 °C with *L. acidophilus* ATCC 4356, *Lm. fermentum* DSM 20052, and *Lc. paracasei* DSM 20312. 

The microbial viability during 28 days of cold storage of these strains is reported in Table 6. During 28 days of cold storage, the cell density of all strains reached values close to 8 log cfu mL^−1^, remaining stable or, in some cases, increasing by one order of magnitude in comparison to the beginning of incubation. 

Here, we underline the ability of *L. acidophilus* ATCC 4356, *L. fermentum* DSM 20052, and *L. paracasei* DSM 20312 to survive and grow in legume-based beverages during cold storage, at viable loads higher than 7 log cfu mL^−1^, respecting the minimum value recommended for the daily intake of lactic acid bacteria throughout these products [42].

In particular, *L. acidophilus* ATCC 4356 and *Lc. paracasei* DSM 20312 showed a significant increase in cell density in both pea and lupin beverages throughout storage, whereas *Lm. fermentum* DSM 20052 showed an increase in viable load followed by a decrease in the late stages of cold storage in both beverages. 

Results of this work are in agreement with those of Liao et al. [43], who reported lactic acid bacteria loads higher than 8 log cfu mL^−1^ in adzuki bean beverages fermented with *Lactococcus lactis* or *Lacticaseibacillus rhamnosus* GG, and stored for 28 days at 4 °C. 

Even though the preparation of legume-based water extracts can be carried out following different protocols, we can conclude that lactic acid bacteria are able to ferment legume extracts and survive in these beverages during cold storage, as also demonstrated for the strains *Lacticaseibacillus rhamnosus* GR-1 and *Streptococcus thermophilus* found in cowpea beverages after 28 days [44]. 

The two-way ANOVA of both lupin and pea grain beverages showed that the cell density was significantly affected by the strain and incubation period, as well as their interaction (*p* ≤ 0.05). The interaction between the strain and incubation period factors significantly influenced the microbial cell densities of *Lactobacillus* strains. 

The high viable cell loads of the three LAB strains determined the decrease in pH values for both the fermented legume-based beverages throughout cold storage. In particular, the pH decreased from 4.81 to 4.00 in lupin-based beverages fermented with *L. acidophilus* and 4.08 to 3.30 in pea-based beverages fermented with *Lc. paracasei*. Despite the high viable concentration found for all strains in both fermented beverages during cold storage, no post-acidification phenomenon was found. This condition is considered useful to preserve both the microbial viability of probiotic strains in foods and the sensory characteristics of the fermented product [45,46,47,48]. During cold storage, the viable cell load increased by one log, on average. The growth of these strains during cold storage, probably resulting from the availability of different types of nutrients [49], was already reported for other lactic acid bacteria strains in different legume-based products [49,50]. 

The lactobacilli strains selected in this work, even though isolated from different matrices, were found to be able to grow in the legume-based water extracts and to survive for one month of cold storage. These results are in agreement with the ability of autochthonous and allochthonous lactic acid bacteria as starters of vegetable matrices [35,51,52] and their survival during cold storage. 

#### 3.3.1. Proteins and Free Peptides/Amino Acids throughout Cold Storage

Table 7 shows the concentration of total protein in fermented legume-based beverages throughout 28 days of cold storage. 

The amount of total free peptides and amino acids remained almost the same during cold storage, as shown in Table 8. Thus, it could be assumed that the water-soluble proteins extracted during the soaking and blending process are a preferential source of organic nitrogen and are metabolized by these strains for their survival. A slight decrease in total protein content during cold storage was found only in beverages inoculated with *L. acidophilus* ATCC 4356 and *L. paracasei* DSM 20312 (Table 7). A limited degree of proteolysis, depending on the strain and fermentation duration, was also reported by Arteaga et al. [53] in lacto-fermented pea protein isolate. On the contrary, Schlegel et al. [54] found that *Limosilactobacillus reuteri* and *Lentilactobacillus parabuchneri* hydrolyzed medium- and low-molecular-weight polypeptides from lupin protein isolate. Based on the extraction and fermentation process here displayed, we speculate that the proteolysis degree of fermented legume grains is the result of the process of protein extraction, the protein profile of the matrix, the LAB strain, and the fermentation conditions.

As already found for total protein content, the concentrations of free peptides and amino acids were significantly affected by the strain but not by the storage period. 

Fermented plant-based foods are often designed as dairy alternatives. However, often, these foods are claimed to contain lower concentrations of nutrients, mainly proteins and peptides, than their milk-based counterparts. It is noteworthy that the concentrations of free peptides and amino acids here reported, and measured by the OPA method, were always higher that those reported by Bhattacharya et al. [55] for 10 commercial dairy products, including milk, ranging from 60 to 130 mg L^−1^, as measured by ion-exchange chromatography. 

#### 3.3.2. Free Amino Acids

The concentrations of amino acids released after the fermentation, as well as those still detectable at the end of cold storage, were quantified by HPLC. Table 9 summarizes the concentrations of different groups of amino acids, which are individually detailed in Appendix A. As already observed for the total concentrations of free peptides and amino acids, the total amount of free amino acids decreased in all fermented legume extracts. The highest reduction in the concentration of total free amino acids was observed in legume grain water extracts fermented with *Lm. fermentum* DSM 20052 (Table 9). As far as the concentration of each amino acid group is concerned, no relevant changes were observed in essential amino acids and ɤ-aminobutyric acid. However, pea water extract fermented with *Lm. fermentum* DSM 20052 showed a great reduction in essential amino acids, branched-chain amino acids, and other amino acids (Table 9). 

As reported in Appendix A, cysteine and tyrosine were largely consumed by all strains, and aspartic acid was reduced preferentially by the *Lm. fermentum* DSM 20052, whereas ɤ-aminobutyric acid remained almost stable in all fermented extracts. In some cases, the reduction in the total content of free amino acids after fermentation was accompanied by an increase in the concentration of certain amino acids. We can speculate that the increased concentration of some amino acids could be related to the reduction in the total protein concentration recorded during cold storage, as reported above (Table 7).

Free amino acids in lupin-based beverages decreased on average by 42% after fermentation. Similar results were also reported for yogurt by Germani et al. [56], who observed a reduction after fermentation of more than 30% in the total amount of free amino acids. 

In the case of pea-based beverages, a sharp reduction in the free amino acid concentration was found only after fermentation with *Lm. fermentum* DSM 20052. It is interesting to note that the free amino acid content of fermented pea beverages is in line with that of cow milk (ca. 450 µmol/L) [57]. It is possible to conclude that, as reported for the total concentration of free peptides and amino acids (Table 8), the concentration of total free amino acids decreased after fermentation but remained almost stable during cold storage (Appendix A).

At the end of the refrigerated period, the total free amino acid content ranged from 223.19 mg L^−1^ to 915.37 mg L^−1^ in lupin-based beverages fermented with *Lm. fermentum* DSM 20052 and pea-based ones fermented with *L. acidophilus* ATCC 4356, respectively (Appendix A). The consumption of fermented legume extracts characterized by a high cell density of probiotic strains and high free amino acid content could improve the concentration of post-prandial blood amino acids. Indeed, Jägeret et al. [58] found that pea extract fermented with the probiotic strains *Lc. paracasei* DSM 20312 and *L. acidophilus* ATCC 4356 increased amino acid absorption after pea protein ingestion. 

In addition, it is interesting to note that a glass (150 mL) of fermented lupin-based beverage contains approximately the same amount of ɤ-aminobutyric acid potentially able to lower systolic blood pressure as demonstrated in humans consuming 50 g per day of GABA-enriched cheese [59].

#### 3.3.3. Sensory Properties

The sensory characteristics of the legume-based beverages were evaluated every seven days during the cold storage period using a simplified check-all-that-apply (CATA) method. Due to the innovative characteristics of these lab-scale fermented beverages and the absence of any similar sensory experience in the untrained panelists, the sensory acceptance of these beverage was considered the main result to be achieved. Thus, questionnaires were compiled describing only a single, the main, trait for each macro-descriptor. The simplified CATA method here applied considered only the acceptability (good/not good) of each descriptor, resulting in a binary response score.

These scores were then organized in a contingency table, combined, and normalized. In some cases, the same descriptor (e.g., acid taste) was considered acceptable (score 1) or not according to the personal preferences of panelists. Here, we considered that scores higher than 0.7 are representative of a sufficient level of acceptability. 

Since there was no information about the sensory descriptors of fermented legume-based beverages in the literature, the descriptors of the CATA method, belonging to macro-descriptors “Appearance”, “Odor”, and “Taste”, were freely defined by each panelist and then compared to each other. Due to the water-like consistency of these beverages, the “Texture” macro-descriptor, necessary to describe yogurt-like fermented beverages, was not included. 

All descriptors provided by panelists for each fermented beverage are reported in Appendix A and are herein summarized in Table 10.

The sensory characteristics of the legume-based beverages were largely and specifically affected by the lactic acid fermentation. In comparison with the unfermented control, lactic acid fermentation of the lupin water extract moderately affected the three sensory macro-descriptors, whereas fermentation of the pea water extract was positively affected by the inoculation of *L. acidophilus* ATCC 4356 and *Lc. paracasei* DSM 20312.

Among the three lactobacilli assayed, *Lm. fermentum* DSM 20052 was the strain that produced limited or no improvements in acceptability scores for all legume-based beverages. None of the three lactobacilli were able to produce a significant improvement in lupin acceptability scores. In this case, the taste was the worst macro-descriptor, since it was characterized by an unpleasant, bitter, and persistent taste in the innermost area of the tongue.

This unacceptable sensory characteristic was reduced thanks to lactic acid fermentation, increasing the taste acceptability from 0 to 0.6. In particular, a milk flavor in samples fermented with *L. acidophilus* ATCC 4356, and vegetable notes in samples fermented with *Lm. fermentum* DSM 20052 and *Lc. paracasei* DSM 20312, partially masked the bitterness (Appendix A). Reduced bitterness was recently found in lupin protein isolates fermented with *Latilactobacillus sakei* subsp. *carnosus* [54].

As far as the appearance of legume-based beverages is concerned, lactic acid fermentation did not negatively affect the appearance of legume samples, with lupin samples characterized by a transparent straw yellow color and with a negligible amount of sediment, and pea samples characterized by a greenish-yellow color (Appendix A). 

All fermented lupin-based beverages were characterized by a cooked ham odor (Appendix A). Similarly, Schlegel et al. [54] found notes of cooked products (cooked potato, roasty, and oatmeal) in lupin fermented with lactic acid bacteria. The best result for the macro-descriptor “Odor” was assigned to pea extracts fermented with *Lc. paracasei* DSM20132, which, as a result of fermentation, produced pleasant notes of “green peas”, and “fruity”, “floral”, and “fresh-cut grass” notes, resulting in a score of 0.9 (Appendix A). These results agree with El Youssef et al. [60], which found “green flavor/vegetal” and “leguminous plant” as the main descriptors of pea protein fermented with lactic acid bacteria. The sensory characteristics of the legume-based beverages showed moderate changes under cold storage, leading to only an increase in odor and taste of acidity (data not shown). 

Our results partially agree with other works in which both extracts of lupin and pea grains were fermented, even though differences in sensory notes described cannot be correctly compared due to differences in fermenting strains, extraction processes, and fermentation steps [61,62]. Even though the method applied is less informative than others based on hedonic scales, it was able to define the level of acceptability of each beverage and underline which sensory trait needs to be improved in order to increase the average level of acceptability.

## 4. Conclusions

This work demonstrates the ability of some strains of lactobacilli to ferment legume water extracts and to survive during cold storage. This result appears particularly interesting in the case of the probiotic strain *L. acidophilus* ATCC 4356. The organoleptic profile of lupin- and pea-based beverages was positively affected by the starter, with the best results obtained with *Lc. paracasei* DSM 20312. Independently of the fermenting strain, high protein and amino acid content was found in lupin-based beverages. In conclusion, the appropriate combination of fermenting strain and legume grains could lead to the production of a legume-based milk substitute containing high concentrations of free peptides and amino acids. Since highly viable lactic acid bacteria were found after up to 28 days of cold storage, these beverages could also be a potential carrier of probiotic lactic acid bacteria. Experiments set up to demonstrate the survival of probiotic microbial cells in the human gastrointestinal tract could offer information about the potential health benefits of these beverages. Moreover, further studies will require a more in-depth beverage sensory characterization in order to improve the preliminary results obtained by the application of the simplified CATA method here employed. 

## Figures and Tables

**Figure 1 foods-11-03346-f001:**
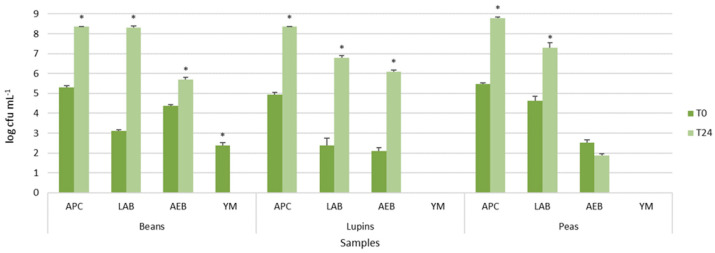
Viable cell counts of main microbial populations of legume grains in water before and after 24 h of incubation at 37 °C. Histograms represent microbial average population values ± standard deviation (error bars). The asterisks indicate statistical differences (*p* < 0.05) after *T* test comparison. Abbreviations: Aerobic Plate Count (APC), Lactic Acid Bacteria (LAB), Aerobic Endospore-Forming Bacteria (AEB), Yeasts and Molds (YM).

**Table 1 foods-11-03346-t001:** *Lactobacillus* spp. strains used for the fermentation of legume-based water extracts [17].

Specie	Strain
*Lactobacillus acidophilus*	ATCC 4356
*Limosilactobacillus fermentum*	DSM 20052
*Lactobacillus gasseri*	ITEM 13541
*Lactobacillus helveticus*	ATCC 15009
*Lactobacillus johnsonii*	NCC533
*Lacticaseibacillus paracaseis* subsp. *paracasei*	DSM 20312
*Lacticaseibacillus rhamnosus*	ATCC 53103

ATCC: American Type Culture Collection, Manassas, Virginia, USA. DSM: Deutsche Sammlung von Mikrorganismen und Zellkulturen GmBH, Braunschweig, Germany. ITEM: Agro-Food Microbial Culture Collection of the Institute of Sciences of Food Production, Bari, Italy. NCC: Nestlé Culture Collection, Lausanne, Switzerland.

**Table 2 foods-11-03346-t002:** Differences in cell densities, reported as Δlog cfu mL^−1^, after 48 h of incubation at 37 °C, for legume grain water extracts inoculated with selected LAB strains.

	Beans	Lupins	Peas	Strain Average Values
*L. acidophilus* ATCC 4356	1.77 ± 0.25 ^Aa^	1.21 ± 0.13 ^Bc^	0.19 ± 0.03 ^Cd^	1.06 ± 0.13
*Lm. fermentum* DSM 20052	1.10 ± 0.10 ^Ab^	3.04 ± 0.10 ^Ba^	1.15 ± 0.07 ^Ac^	1.77 ± 0.09
*L. gasseri* ITEM 13541	0.76 ± 0.10 ^Bc^	0.16 ± 0.08 ^Ce^	0.96 ± 0.06 ^Ac^	0.63 ± 0.08
*L. helveticus* ATCC 15009	0.31 ± 0.03 ^Bd^	0.04 ± 0.02 ^Ce^	2.52 ± 0.10 ^Aa^	0.96 ± 0.05
*L. johnsonii* NCC533	1.06 ± 0.06 ^Ab^	0.74 ± 0.11 ^Bd^	−0.40 ± 0.13 ^Ce^	0.47 ± 0.10
*Lc. paracasei* DSM 20312	1.64 ± 0.26 ^Aa^	1.54 ± 0.22 ^Ab^	1.66 ± 0.12 ^Ab^	1.61 ± 0.2
*Lc. rhamnosus* ATCC 53103	0.32 ± 0.19 ^Bd^	0.88 ± 0.15 ^Ad^	−0.25 ± 0.08 ^Ce^	0.32 ± 0.14

Two-way ANOVA analysis was applied to estimate the effect of legume and starter strain on cell density. The least significant difference comparison values (LSD, 95% confidence interval) were calculated. Cell densities: legume, 0.16 Δlog cfu g^−1^; strain, 0.25 Δlog cfu g^−1^. Superscript letters indicate significant differences within rows, while lowercase letters indicate significant differences within columns.

**Table 3 foods-11-03346-t003:** Differences in pH values, reported as ΔpH values, after 48 h of incubation at 37 °C, for legume grain water extracts inoculated with selected LAB strains.

	Beans	Lupins	Peas	Strain Average Values
*L. acidophilus* ATCC 4356	−2.58 ± 0.20 ^Bd^	−0.58 ± 0.16 ^Aa^	−3.57 ± 0.08 ^Cc^	−2.24 ± 0.14
*Lm. fermentum* DSM 20052	−1.51 ± 0.06 ^Bb^	−1.12 ± 0.14 ^Ab^	−3.51 ± 0.21 ^Cc^	−2.05 ± 0.14
*L. gasseri* ITEM 13541	−1.48 ± 0.19 ^Ab^	−1.72 ± 0.17 ^Bc^	−3.24 ± 0.12 ^Cb^	−2.15 ± 0.16
*L. helveticus* ATCC 15009	−1.81 ± 0.24 ^Bc^	−1.17 ± 0.07 ^Ab^	−3.44 ± 0.11 ^Cc^	−2.14 ± 0.14
*L. johnsonii* NCC533	−1.56 ± 0.22 ^Ab^	−2.06 ± 0.15 ^Bd^	−3.02 ± 0.11 ^Cb^	−2.21 ± 0.16
*Lc. paracasei* DSM 20312	−1.56 ± 0.22 ^Bb^	−1.24 ± 0.14 ^Ab^	−3.67 ± 0.09 ^Cc^	−2.15 ± 0.15
*Lc. rhamnosus* ATCC 53103	−0.59 ± 0.14 ^Aa^	−0.98 ± 0.05 ^Bb^	−2.12 ± 0.18 ^Ca^	−1.23 ± 0.12

Two-way ANOVA analysis was applied to estimate the effect of legume and starter strain on pH values. The least significant difference comparison values (LSD, 95% confidence interval) were calculated. pH values: legume 0.19 ΔpH; strain, 0.29 ΔpH. Superscript letters indicate significant differences within rows, while lowercase letters indicate significant differences within columns. Initial pH values: beans 6.62, lupins 5.78, peas 7.02. Negative values represent the unit of pH reduction of fresh extracts that occurred during fermentation.

**Table 4 foods-11-03346-t004:** Differences in cell densities, reported as Δlog cfu mL^−1^ after 24 and 48 h of incubation at 37 °C, of legume grain water extracts inoculated with selected LAB strains.

	Beans		Lupins		Peas	
	t24	t48	t24	t48	t24	t48
*L. acidophilus* ATCC 4356	2.81 ± 0.21 ^Aa^	3.24 ± 0.07 ^Aa^	2.76 ± 0.09 ^Aa^	3.15 ± 0.06 ^Aa^	1.87 ± 0.02 ^Ba^	2.42 ± 0.07 ^Ba^
*Lm. fermentum* DSM 20052	2.97 ± 0.04 ^Aa^	3.43 ± 0.06 ^Aa^	2.23 ± 0.10 ^Bb^	2.64 ± 0.21 ^Bb^	1.85 ± 0.07 ^Ca^	2.63 ± 0.11 ^Ba^
*Lc. paracasei* DSM 20312	2.17 ± 0.07 ^Bb^	3.19 ± 0.11 ^Ab^	2.89 ± 0.10 ^Aa^	3.08 ± 0.07 ^Aa^	1.68 ± 0.05 ^Ca^	2.18 ± 0.11 ^Bb^

Two-way ANOVA analysis was applied to estimate the effect of legume and starter strain on cell density. The least significant difference comparison values (LSD, 95% confidence interval) were calculated. Cell densities: legume, 0.19 Δlog cfu g^−1^; strain, 0.21 Δlog cfu g^−1^. At each sampling time, superscript letters indicate significant differences within rows, while lowercase letters indicate significant differences within columns.

**Table 5 foods-11-03346-t005:** Differences in pH values, reported as ΔpH values after 24 and 48 h of incubation at 37 °C, of legume grain water extracts inoculated with selected LAB strains.

	Beans		Lupins		Peas	
	t24	t48	t24	t48	t24	t48
*L. acidophilus* ATCC 4356	−1.01 ± 0.12 ^Bc^	−2.19 ± 0.10 ^Bc^	−0.28 ± 0.06 ^Aa^	−1.19± 0.04 ^Aa^	−2.45 ± 0.05 ^Cc^	−3.42 ± 0.17 ^Cc^
*Lm. fermentum* DSM 20052	−0.76 ± 0.11 ^Bb^	−0.56 ± 1.79 ^Aa^	−0.95 ± 0.07 ^Cb^	−1.15 ± 0.06 ^Ba^	−0.46 ± 0.04 ^Aa^	−2.86 ± 0.06 ^Ca^
*Lc. paracasei* DSM 20312	−0.16 ± 0.05 ^Aa^	−1.87 ± 0.02 ^Bb^	−0.38 ± 0.03 ^Ba^	−1.14 ± 0.06 ^Aa^	−1.01 ± 0.12 ^Cb^	−3.05 ± 0.07 ^Cb^

Two-way ANOVA analysis was applied to estimate the effect of legume and starter strain on pH values. The least significant difference comparison values (LSD, 95% confidence interval) were calculated. pH values: legume 0.15 ΔpH; strain, 0.16 ΔpH. At each sampling time, superscript letters indicate significant differences within rows, while lowercase letters indicate significant differences within columns. Initial pH values: beans 6.59, lupins 5.81, peas 6.98. Negative values represent the unit of pH reduction of fresh extracts after fermentation.

**Table 6 foods-11-03346-t006:** Cell density (log cfu mL^−1^) of selected lactobacilli in legume-based beverages during 28 days of storage at 4 °C.

Beverage	Days of Storage	*L. acidophilus* ATCC 4356	*Lm. fermentum* DSM 20052	*Lc. paracasei* DSM 20312
Lupins	0	7.60 ± 0.06 ^Bd^	7.57 ± 0.07 ^Be^	7.72 ± 0.05 ^Ac^
	7	8.40 ± 0.05 ^Bb^	7.71 ± 0.07 ^Cd^	8.63 ± 0.06 ^Ab^
	14	8.11 ± 0.04 ^Cc^	8.92 ± 0.06 ^Aa^	8.69 ± 0.04 ^Bb^
	21	8.83 ± 0.06 ^Aa^	7.89 ± 0.09 ^Bc^	8.80 ± 0.10 ^Aa^
	28	8.77 ± 0.07 ^Aa^	8.30 ± 0.06 ^Bb^	8.86 ± 0.07 ^Aa^
	Average values	8.67 ± 0.08	8.36 ± 0.05	8.65 ± 0.06
Peas	0	7.54 ± 0.08 ^Bd^	7.59 ± 0.06 ^Ac^	7.67 ± 0.08 ^Ac^
	7	7.87 ± 0.07 ^Ac^	7.76 ± 0.09 ^Ab^	7.66 ± 0.07 ^Bc^
	14	8.06 ± 0.06 ^Ab^	7.72 ± 0.13 ^Bb^	7.92 ± 0.04 ^Bb^
	21	8.07 ± 0.05 ^Bb^	8.06 ± 0.07 ^Ba^	8.33 ± 0.09 ^Aa^
	28	8.55 ± 0.06 ^Aa^	7.54 ± 0.08 ^Cc^	8.39 ± 0.07 ^Ba^
	Average values	8.17 ± 0.04	7.76 ± 0.08	8.09 ± 0.08

Two-way ANOVA analysis was applied to estimate the effect of time of cold storage and starter strain on cell density values. The least significant difference comparison values (LSD, 95% confidence interval) were calculated for each factor. Lupin beverage: time, 0.12 log cfu g^−1^; strain, 0.10 log cfu g^−1^. Pea beverage: time, 0.14 log cfu g^−1^; strain, 0.11 log cfu g^−1^. Superscript letters indicate significant differences within rows, while lowercase letters indicate significant differences within columns.

**Table 7 foods-11-03346-t007:** Concentration of total protein (mg mL^−1^) of fermented beverages and in unfermented water extracts during 28 days of refrigerated (4 °C) storage.

Beverage	Days of Cold Storage	Unfermented Extract	Fermented Beverages		
			*L. acidophilus* ATCC 4356	*Lm. fermentum* DSM 20052	*Lc. paracasei* DSM 20312
Lupins	0	1.22 ± 0.09 ^Ab^	0.78 ± 0.17 ^Ba^	0.97 ± 0.09 ^Ba^	1.05 ± 0.06 ^Aa^
	7	1.58 ± 0.05 ^Aa^	0.60 ± 0.26 ^Ba^	0.78 ± 0.17 ^Ba^	0.66 ± 0.08 ^Bb^
	14	1.79 ± 0.05 ^Aa^	0.66 ± 0.37 ^Ca^	0.96 ± 0.23 ^Ba^	0.76 ± 0.10 ^Bb^
	21	1.66 ± 0.06 ^Aa^	0.54 ± 0.26 ^Ca^	0.82 ± 0.10 ^Ba^	0.74 ± 0.07 ^Bb^
	28	1.27 ± 0.05 ^Ab^	0.34 ± 0.37 ^Cb^	0.73 ± 0.18 ^Bb^	0.51 ± 0.05 ^Bc^
Peas	0	0.26 ± 0.04 ^Ab^	0.05 ± 0.04 ^Ca^	0.06 ± 0.04 ^Bc^	0.12 ± 0.03 ^Ba^
	7	0.24 ± 0.04 ^Ac^	0.06 ± 0.04 ^Ba^	0.05 ± 0.05 ^Bc^	0.08 ± 0.02 ^Ba^
	14	0.24 ± 0.07 ^Ac^	0.04 ± 0.03 ^Ba^	0.05 ± 0.04 ^Bc^	0.07 ± 0.03 ^Ba^
	21	0.35 ± 0.05 ^Ab^	nd	0.14 ± 0.05 ^Bb^	0.07 ± 0.04 ^Ca^
	28	0.50 ± 0.07 ^Aa^	nd	0.25 ± 0.06 ^Ba^	0.07 ± 0.05 ^Ca^

Two-way ANOVA analysis was applied to estimate the effect of time of cold storage and starter strain on total protein values. The least significant difference comparison values (LSD, 95% confidence interval) were calculated for each factor. Lupin beverage: time, 0.25 mg mL^−1^; strain, 0.22 mg mL^−1^. Pea beverage: time, 0.07 mg mL^−1^; strain, 0.06 mg mL^−1^. Superscript letters indicate significant differences within rows, while lowercase letters indicate significant differences within columns. nd = not detected.

**Table 8 foods-11-03346-t008:** Concentration of free peptides and amino acids (mg mL^−1^) in fermented beverages and in unfermented water extracts during 28 days of refrigerated (4 °C) storage.

Beverage	Days of Cold Storage	Unfermented Extract	Fermented Beverages		
			*L. acidophilus* ATCC 4356	*Lm. fermentum* DSM 20052	*Lc. paracasei* DSM 20312
Lupin	0	3.94 ± 0.05 ^Ba^	3.56 ± 0.09 ^Ba^	4.72 ± 0.04 ^Aa^	3.43 ± 0.03 ^Ba^
	7	4.31 ± 0.07 ^Aa^	2.82 ± 0.48 ^Cb^	3.65 ± 0.05 ^Bb^	3.00 ± 0.04 ^Ca^
	14	4.16 ± 0.14 ^Aa^	2.75 ± 0.65 ^Bb^	3.86 ± 0.07 ^Ab^	2.91 ± 0.07 ^Ba^
	21	3.63 ± 0.06 ^Ab^	2.80 ± 0.55 ^Bb^	3.75 ± 0.06 ^Ab^	2.63 ± 0.01 ^Bb^
	28	3.39 ± 0.15 ^Ab^	2.52 ± 1.17 ^Bb^	4.51 ± 0.02 ^Aa^	2.85 ± 0.02 ^Bb^
Pea	0	3.26 ± 0.05 ^Ab^	2.26 ± 0.06 ^Cd^	3.22 ± 0.05 ^Aa^	2.76 ± 0.03 ^Bc^
	7	3.34 ± 0.04 ^Ab^	2.63 ± 0.04 ^Bb^	2.58 ± 0.09 ^Bc^	2.55 ± 0.08 ^Bd^
	14	3.47 ± 0.02 ^Aa^	3.09 ± 0.10 ^Ba^	2.77 ± 0.11 ^Db^	2.95 ± 0.04 ^Cb^
	21	3.53 ± 0.06 ^Aa^	2.55 ± 0.05 ^Dc^	2.72 ± 0.05 ^Cb^	3.43 ± 0.03 ^Ba^
	28	3.05 ± 0.05 ^Ac^	2.72 ± 0.08 ^Cb^	2.84 ± 0.07 ^Bb^	3.00 ± 0.04 ^Ab^

Two-way ANOVA analysis was applied to estimate the effect of time of cold storage and starter strain on total free peptide and amino acid values. The least significant difference comparison values (LSD, 95% confidence interval) were calculated for each factor. Lupin beverage: time, 0.57 mg mL^−1^; strain, 0.51 mg mL^−1^. Pea beverage: time, 0.10 mg mL^−1^; strain, 0.09 mg mL^−1^. Superscript letters indicate significant differences within rows, while lowercase letters indicate significant differences within columns.

**Table 9 foods-11-03346-t009:** Concentrations of free amino acids (expressed in mg L^−1^) in fermented beverages and in unfermented water extracts at the beginning (T0) and at the end (T28) of refrigerated (4 °C) storage period.

Beverage	Groups of Amino Acids	Unfermented Control		Fermented Beverages					
				*L. acidophilus* ATCC 4356		*Lm. Fermentum* DSM 20052		*Lc. paracasei* DSM 20312	
		T0	T28	T0	T28	T0	T28	T0	T28
Lupin	Essential amino acids ^1^	53.93	48.72	23.80	20.21	16.54	13.05	19.56	17.35
	Br. chain amino acids ^2^	72.48	69.79	16.17	12.84	8.15	6.59	2.11	1.51
	Other amino acids	528.43	494.60	404.68	331.77	265.70	203.55	378.34	326.65
	ɤ-aminobutyric acid	106.64	105.93	105.27	96.95	101.55	97.55	104.54	102.92
	Total amount	761.48	719.04	549.92	461.77	391.94	320.74	504.55	448.43
Pea	Essential amino acids ^1^	107.83	105.53	108.65	103.06	64.25	50.62	112.22	99.01
	Br. chain amino acids ^2^	106.21	106.49	120.92	101.97	91.66	77.29	121.14	107.48
	Other amino acids	753.18	760.11	738.47	710.34	629.57	513.56	756.34	691.66
	ɤ-aminobutyric acid	40.37	38.06	45.48	41.15	45.35	39.57	45.90	38.61
	Total amount	1007.59	1010.19	1013.52	956.52	830.83	681.04	1035.6	936.76

^1^, Thr + Met + Phe + Trp; ^2^, Val + Leu + Ile.

**Table 10 foods-11-03346-t010:** Comparison between acceptability scores of macro-sensory descriptors of fermented legume grain beverages and unfermented water extracts.

Beverage	Macro-Descriptors	Water Extract	Fermented Beverages		
			*L. acidophilus* ATCC 4356	*Lm. fermentum* DSM 20052	*Lc. paracasei* DSM 20312
Lupin	Appearance	0.7	0.6	0.6	0.6
	Odor	0.8	0.7	0.5	0.7
	Taste	0.0	0.6	0.4	0.6
Pea	Appearance	0.4	0.5	0.3	0.5
	Odor	0.8	0.8	0.4	0.9
	Taste	0.7	0.7	0.7	0.9

## Data Availability

The datasets generated for this study are available on request to the corresponding author.

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
