# Peer review of "Use of Selected Lactic Acid Bacteria for the Fermentation of Legume-Based Water Extracts"

_foods, 2022, doi:10.3390/foods11213346_

Round 1
Reviewer 1 Report (Previous Reviewer 1)
The manuscript has been improved at the writing level, but the experimental part must be improved.
Author Response
"Please see the attachment."

Reviewer 2 Report (Previous Reviewer 2)
Dear Editor (s) and Authors,
Despite this third revision of the manuscript (Use of selected lactic acid bacteria for the fermentation of legume-based water extracts), the authors did not answer some important questions.
1-The fermentation process of the aqueous extracts of legumes was at a temperature of 37°C. Why did the authors use a temperature of 30°C when calculating the numbers of lactic acid bacteria?
The optimum temperature for the growth of Probiotic bacteria used in fermentation is 37 °C, which is similar to the temperature of the human body.
Why did the authors use a method of work or a scientific source that did not fit the research plan?
There are many scientific references that indicate that the best incubation temperature for probiotic bacteria is 37°C.
Further input is required prior to consideration for accepting the manuscript. Every published article might become a model for other (young) researchers or become a reference for further studies.
2-In the previous evaluation of the manuscript, I indicated that the names of lactic acid bacteria should be written according to the modern nomenclature, but there are some names that are still in the old nomenclature. see page 8 line 320 and line 324, Correct Lactobacillus rhamnosus to Lacticaseibacillus rhamnosus.
3-The authors indicated that the unit of measurement for the numbers of microorganisms is cfu. ml-1, but we note in the results the authors write cfu/ml!!!!. see page 3 line 105 and page 8.
4-All units of measurement in the manuscript must be corrected such as mgL-1 correct to mg.L-1.

Author Response
"Please see the attachment."

Reviewer 3 Report (New Reviewer)
This study is relevant and interesting. The manuscript is well organized and well written. However some typosettings should be control. As I have mentioned in the pdf.file of the manuscript;
1. Please revise all "cfu. mL-1" carefully.
2. Line 356. I think that it is a subtitle.
3. Please revise the citations in Lines 318, 376, 400, 440, 499 and 504 as for example Liao et al [43].
4. Line 332-333 Please choose to write data as 4.81 to 4.00 ........ 4.08 to 3.30

Author Response
"Please see the attachment."

Round 2
Reviewer 2 Report (Previous Reviewer 2)
Dear Editors,
Authors did all necessary changes to improve the manuscript and now I recommend it for publication in the current form.
Best.
Author Response
Dear Reviewer 2,
thanks for your positive respons to our manuscript.
Here upload the final version after changes requested by the Academic Editor.
Best regrads
F. Baruzzi

This manuscript is a resubmission of an earlier submission. The following is a list of the peer review reports and author responses from that submission.
Round 1
Reviewer 1 Report
The manuscript details an interesting topic, but the data is considered preliminary in nature. Probiotic viability of the plant-based beverages was above the minimum recommended after 28 days, and the nutritional properties may be improved but further studies are required to know if the microorganisms are able to survive during gastrointestinal digestion in order to reach the colon to exert their beneficial effects on human gut health.
Abstract: to check the number of words, maximum 200.
Line 32: separation of keywords using ; not ,
Lines 42, 58, 94, etc.: to check the space between words throughout the manuscript.
Lines 47 to 51: the positive effects/advantages of producing or reducing the concentration of these compounds by LAB should be explained.
Line 52: Glycine max in cursive.
Line 70 to 73: to include the reference.
Line 75: Acetic acid may contribute to a lower acceptance of the taste of fermented beverages, being detected organoleptically at 0.40 g/Litre (Siebert KJ. 1999. Modeling the flavor thresholds of organic acids in beer as a function of their molecular properties. Food Qual Preference. 10(2):129–137).
Line 80: Authors tended to on occasion use sweeping statements such as on L 80, L43, L321… using “can”. I do not believe that there is sufficient evidence to support such a claim and some modifying words such as “could” “may” …. should be included throughout the manuscript.
Lines 96, 98, 111, 116, etc: to check the space between the units and number: h, °C, throughout the manuscript.
Line 97, etc: to check capital letters like PCA: Plate Count Agar, throughout the manuscript.
Line 105: to define the units “cfu” previously.
Line: 114: authors should mention information of the centrifuge, model, country, etc.
Figure 1: to indicate in the figure caption what the error bars indicate, and it is necessary to add “before and after 24h…. The statistical analysis should be added.
Lines 204, 205: to indicate previously Lm., Leuc.
As suggested in line 210, the legume-based aqueous extracts must be pasteurized to avoid microbiological contamination. In addition, bean extracts contained molds and yeasts. The fermentation must be done after the heat treatment, since when adding new Lactobacillus strains, authors did not indicate or distinguish whether the growth obtained could be also due to the autochthonous bacteria as LAB of the aqueous extracts.
Table 2 and 3: The statistical analysis should be added, and negative values were not clear, it is better to add initial and final pH values and then the difference between them (panel b).
Line 241 to 243: What samples were those values from?
Line 248 to 249: Why the acidification rate was not considered to be unpleasant?
From line 263, Part 3.3: Only the results were presented, the discussion needs to be improved
Table 4: The statistical analysis should be added. Lines 297 and 298: What does that information mean?
Tables 5, 6, 7: The statistical analysis should be added.
From line 321: What data do you base on to confirm that statement?
From line 330 to 333: Why were the reported values higher than other studies? It should be discussed.
Regarding sensory analysis (part 2.3.3. and 3.3.3.), the method has not been carried out correctly, then the results obtained are not valid. First of all, the Check-All-That-Apply (CATA) methodology is used for consumer studies. In these studies, the consumer is asked how much he or she likes or dislikes a product with respect to various product attributes. To obtain valid results, more than 100 consumers are necessary. Besides, the scale measurement is never from 0 to 1, but from 1 to 9 (being 5, I neither like nor dislike it), because the inexperienced consumer cannot assess a product in such a short range (0-1).
Secondly, there are descriptive tests, these tests could be carried out with 10 panelists, but in this study, the judges are not trained, and for a descriptive test this is mandatory. To carry out correctly a descriptive analysis of a product, the judges are trained for 5 or more sessions, so that they are experts in that product and thus be able to judge it. Once the training is done, the judges who have not verified their validity as panelists are discarded. Thus, in a quantitative way, the attributes of the product are valued through a specific lexicon for the product and developed in the training sessions. In a descriptive analysis appearance, flavor and aroma are not only judged, but the appearance, flavor and aroma are deepened, differentiating each attribute, not making an average of the score. Therefore, each attribute must be scored individually.
It should be noted that this study should add the category of texture in sensory analysis.
Table S4: what does the natural attribute in appearance mean?
Finally, I recommend checking the style in the references, because the authors use capital letters in the title in some references (3, 11, 15,17, 37)
Reviewer 2 Report
Dear Editors and Authors,
The manuscript (Use of selected lactic acid bacteria for the fermentation of legume-based water extracts) needs many corrections and modifications:
1-The abstract of the manuscript needs to add some results as numbers in order to be clear to the reader.
2- Scientific names of bacteria should be written in italics throughout the manuscript, see page 2 line 60, page 3 line 108,.......etc.
3- Many of the working methods do not contain scientific references. For example, the method of preparing aqueous extracts, the method of calculating bacterial numbers, and others, I suggested adding some references, please, see the attached file.
4- Page 2 line 98, Why used 30 C of total aerobic bacteria? ????
5-Page 3 line 99, The Man, Rogosa and Sharpe media used to count lactobacilli in the lactic acid bacteria group only, sub-headings in the manuscript should be changed to count lactobacilli numbers. This culture media is not used to calculate all types of lactic acid bacteria, only Lactobacilli group.
6-Page 4 line 185, MRS media used in calculating the numbers of lactic acid bacteria, it is a selective media for lactobacilli bacteria, how were the numbers calculated for cocci bacteria belonging to the same group.
7-Discussing some results you need to add references and compare with their results, I suggested adding (Page 5 line 204, Al-Sahlany, S. T., & Niamah, A. K. (2022). Bacterial viability, antioxidant stability, antimutagenicity and sensory properties of onion types fermentation by using probiotic starter during storage. Nutrition & Food Science.) page 5 line 211, UKMAWATI, S., ROSALINA, F., SIPRIYADI, S., DEWI, N. K., YUNITA, M., SARHAN, A. R. T., ... & KUSUMAWATI, E. (2022). Bacterial diversity of mangrove ecosystem in Klawalu Sorong, West Papua, Indonesia. Biodiversitas Journal of Biological Diversity, 23(3).
8-Table 2 and 3, What does Δlog cfu mL-1 and ΔpH mean is not clear, nothing is mentioned about these two terms in the working methods, why and how according to the difference between the coefficients? everything is not clear.
9-Table 4, How did the numbers of bacteria increase logarithmic cycle during the storage period? While these bacteria such as Lactobacillus acidophilus and Lm. fermentum cannot grow at 15°C, there must be a clear and scientific explanation.
10- Figure 1 , The results of calculating the numbers of yeasts and molds must be added in the figure with the rest of the results, to be clear to the reader.
11-Table 6, There is no clear explanation for the result of the decrease in the proportion of peptides after 14 days of storage, scientifically it is supposed to increase with the increase of microorganisms Table 2 and the length of the storage period.
12- The conclusions in the manuscript contain more results than the abstract, and they must be rewritten and the results removed from it.
